# Generalising the HCP multimodal cortical parcellation to UK Biobank using geometric deep learning

**Logan Z. J. Williams**[1,2,3]                                    LOGAN.WILLIAMS@KCL.AC.UK
[1] *Research Department of Biomedical Computing, School of Biomedical Engineering and Imaging Sciences, King's College London, London, SE1 7EH, UK*
[2] *Centre for the Developing Brain, School of Biomedical Engineering and Imaging Science, King's College London, London, SE1 7EH, UK*
[3] *Starship Children's Hospital, Auckland, 1023, New Zealand*

**Simon Dahan**[1]                                                SIMON.DAHAN@KCL.AC.UK
**Tanya Poppe**[4]                                                TANYAPOPPE@GMAIL.COM
[4] *Aotearoa Clinical Trials, Middlemore Hospital, Auckland, 2025, New Zealand*

**Yourong Guo**[1,2]                                              YOURONG.GUO@KCL.AC.UK
**Matthew F. Glasser**[5,6]                                       GLASSERM@WUSTL.EDU
[5] *Department of Neuroscience, Washington University Medical School, Saint Louis, 63110, Missouri, USA*
[6] *Department of Radiology, Washington University Medical School, Saint Louis, 63110, Missouri, USA*

**Fidel Alfaro-Almagro**[7]                                      FIDEL.ALFAROALMAGRO@NDCN.OX.AC.UK
[7] *Wellcome Centre for Integrative Neuroimaging, Nuffield Department of Clinical Neurosciences, University of Oxford, Oxford, OX3 9DU, UK*

**Mohamed A. Suliman**[1]                                        MOHAMED.SULIMAN@KCL.AC.UK
**Anderson M. Winkler**[8]                                       ANDERSONWINKLER@GMAIL.COM
[8] *Department of Human Genetics, University of Texas Rio Grande Valley, Brownsville, 78520, Texas, USA*

**Timothy S. Coalson**[5,6]                                      COALSONT@WUSTL.EDU
**David C. Van Essen**[5,6]                                      VANESSEN@WUSTL.EDU
**Stephen M. Smith**[7]                                          STEPHEN.SMITH@NDCN.OX.AC.UK
**Emma C. Robinson**[1,2]                                        EMMA.ROBINSON@KCL.AC.UK

**Editors:** Under Review for MIDL 2026

## Abstract

The Human Connectome Project Multimodal Parcellation (HCP_MMP1.0) provides a robust *in vivo* map of the cerebral cortex, which demonstrates variability in structure and function that cannot be captured through diffeomorphic image registration alone. The HCP successfully employed a fully-connected neural network architecture to capture this variation, however it is unclear whether this approach generalises to other datasets with less rich imaging protocols. In this paper we propose and validate a novel geometric deep learning framework for generating individualised HCP_MMP1.0 parcellations in UK Biobank (UKB), an extremely rich resource that has led to numerous breakthroughs in neuroscience. To address substantial differences in image acquisition (for example, 6 minutes of resting-state fMRI per subject for UKB vs. 60 minutes per subject for HCP), we introduce a multi-step learning procedure including pretraining with a convolutional autoencoder. Compared to a fully-connected baseline, our proposed framework improved average detection rate across all areas by 10.4% (99.1% vs 88.7%), and detection of the worst performing area by 51.0% (80.9% vs. 29.9%). Importantly, this was not a result of the framework predicting one consistent parcellation across subjects, as visual inspection indicated that our method was sensitive to atypical cortical topographies.

**Keywords:** Geometric deep learning, image segmentation, UK Biobank, Human Connectome Project, cortical surfaces

## 1. Introduction

The Human Connectome Project Multimodal Parcellation (HCP_MMP1.0) provides a robust *in vivo* map of the human cerebral cortex (Glasser et al., 2016a,b), which has been implicated in cognition, behaviour and neuropsychiatric disorders (Sydnor et al., 2021). Using an observer-independent approach and multimodal cortical surface registration, the HCP identified 83 areas previously reported using post-mortem histology, and 97 new areas defined by sharp transitions in structure and function (Glasser et al., 2016a). Importantly, these cortical areas demonstrated marked diversity in structural and functional organisation that cannot be captured through biomechanically-constrained, diffeomorphic registration alone (Figure 1) (Glasser et al., 2016a). To generate parcellations that reflected individual differences in cortical organisation, the HCP used 112 unique features of cortical structure and function to train a fully-connected neural network (FCNN) that accurately predicted which cortical area each vertex belonged to (360 networks in total), even when they significantly differed from the group average. These individualised HCP_MMP1.0 parcellations, which have significantly advanced our understanding of brain-behaviour relationships (Bijsterbosch et al., 2018) and cognitive function (Assem et al., 2020), could benefit other large-scale neuroimaging consortia. One such study is the UK Biobank (UKB), an extremely rich resource that has led to a number of neuroscientific breakthroughs (Miller et al., 2016; Elliott et al., 2018; Wang et al., 2022). However, as these parcellations utilised large amounts of high-resolution MRI data per subject (including an extensive task fMRI battery) (Glasser et al., 2016b), it is unknown whether individualised, high-quality parcellations are also achievable in studies that cannot match these protocols (Glasser et al., 2016b; Bijsterbosch et al., 2020).

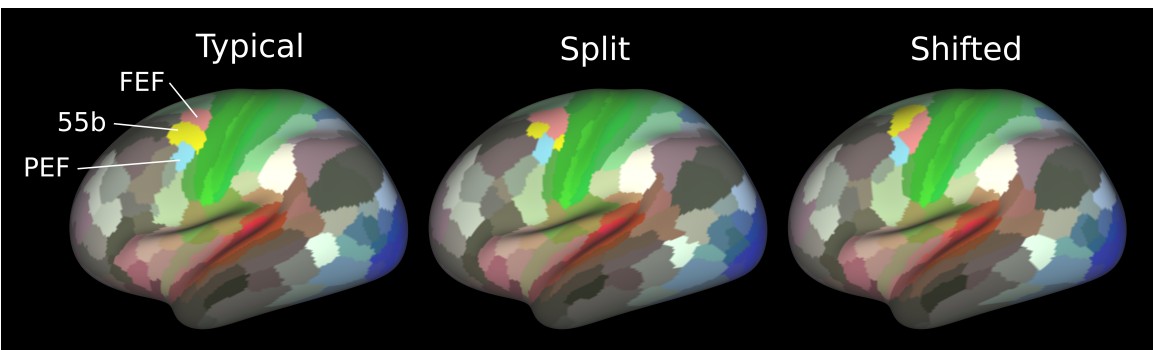

Figure 1: Representative examples of typical, split and shifted area 55b topographies in individual HCP subjects, generated using the proposed geometric deep learning framework. Inter-individual differences reflect variabilities that persist following diffeomorphic multimodal alignment to a single group average template.

The HCP FCNN approach demonstrated overall excellent performance, detecting 96.6% of all cortical areas in 210 unseen subjects. However, some areas were detected <75% of the time (Glasser et al., 2016a). It is unknown whether these areas are truly absent, or whether this reflects inherent limitations in FCNN architectures, namely that each vertex

is treated as spatially independent of its neighbours. Convolutional neural networks are one of the most successful learning-based methods for semantic segmentation, and operate by passing localised filters over an image to capture context-specific information (LeCun et al., 2015). This work introduces and validates a new geometric deep learning (gDL) framework for generating individualised HCP_MMP1.0 parcellations on sphericalised cortical surfaces. Specifically, this approach: 1) improves areal detection rates compared to a FCNN baseline; 2) remains sensitive to individual topographic variations in areal organisation that cannot be captured through diffeomorphic image registration alone; and 3) generalises the HCP_MMP1.0 parcellation to a new dataset with a substantially different image acquisition protocol.

## 2. Materials and Methods

### 2.0.1. Data

HCP images were acquired with a customized 3T Siemens 'Connectom' Skyra scanner and standard Siemens 32-channel head coil at a single site in Washington University, St. Louis (Glasser et al., 2013). Training examples were taken from the same 210 HCP subjects ('210P'; 29.4 ± 3.5 years, 130 biological females (61.9%)) used to develop the original HCP MMP_1.0 (Glasser et al., 2016a). HCP image acquisition consisted of structural: 0.7mm isotropic T1w and T2w; resting-state fMRI (rs-fMRI): 2.0mm isotropic, 60 minutes total; task-fMRI: 7 tasks, 48 minutes total (Glasser et al., 2013).

Two cohorts were used for validation: 1) a separate set of 210 HCP subjects ('210V'; 28.8 ± 3.5 years, 114 biological females (54.3%), same as used by (Glasser et al., 2016a)) and 2) 1500 UKB subjects (63.8 ± 7.6 years, 822 biological females (54.8%)). UKB images were acquired with a standard 3T Siemens Skyra scanner and Siemens 32-channel head coil at 4 sites across England (Alfaro-Almagro et al., 2018). UKB image acquisition consisted of structural: 1mm isotropic T1w and T2-FLAIR; rs-fMRI: 2.4mm isotropic, 6 minutes total; task-fMRI: 1 task, 4 minutes total (Alfaro-Almagro et al., 2018).

### 2.0.2. Pre-processing

Major differences in scanning protocols exist between HCP and UKB datasets, with UKB acquiring lower resolution structural and functional imaging, and substantially less fMRI per subject (fewer tasks and 6 vs. 60 mins of rs-fMRI). This necessitated the development and validation of new tools for generalising the HCP_MMP1.0 to UKB data. To achieve this we first ran the HCP minimal preprocessing pipeline (Glasser et al., 2013) to allow multimodal surface-based registration (Robinson et al., 2014, 2018) of individual subject cortical features to the HCP MSMAll template space. This drives alignment using features on sphericalised cortical surfaces that better correspond with cortical areas, specifically: 1 T1w/T2w ratio ('myelin') map, 32 rs-fMRI ICA spatial maps and 9 visuotopic spatial maps) (Glasser et al., 2016a). This required generating T1w/T2w ratio maps using the intensity bias correction described in (Glasser et al., 2013), mapping the T1w/T2w ratio and denoised rs-fMRI timeseries to the cortical surface, and generating individualised rs-fMRI ICA spatial

maps through weighted dual regression of the HCP group ICA spatial template[1] into UKB rs-fMRI timeseries data (Glasser et al., 2016a). Compared to volumetric and folding-based surface registration, this multimodal surface registration approach markedly improves inter-individual alignment of cortical areas (Glasser et al., 2016a; Coalson et al., 2018; Robinson et al., 2014, 2018).

### 2.0.3. Model design

Parcellation of the cerebral cortex was framed as a binary classification problem, where models were tasked with labelling vertices as being part of an area or not (Glasser et al., 2016a). Model inputs were 91 cortical features including cortical thickness, curvature, T1w/T2w ratio, 77 rs-fMRI ICA spatial maps, 6 visuotopic spatial maps and 5 artefact maps[2]. Vertex classification was spatially constrained to occur within a restricted region (a 'searchlight' containing vertices within 30mm geodesic distance of the group average area). This was possible because all data had been multimodally aligned during pre-processing and justified since the relative positions of cortical areas are strongly conserved (Krubitzer, 2007). In total 360 classification networks were trained, one for each area. Areas from left and right hemispheres were considered separately, since some areas exhibit important asymmetries in their functional connectivity and/or spatial relationships with adjacent areas (Glasser et al., 2016a).

**Fully-connected framework:** An optimised FCNN architecture, which better reflects current practices, was used as a baseline to ensure a fair comparison with the gDL framework. Our optimised FCNN architecture differed from the original HCP FCNN (Glasser et al., 2016a) in the following ways: 1) the number of hidden layers was increased from one $(H_1 = 9)$ to three, with dimensions of 128 $(H_1)$, 64 $(H_2)$ and 32 $(H_3)$; 2) tahn non-linearities were replaced with SiLU non-linearities; 3) batchnorm was used after each non-linearity; 4) the mean squared error loss was replaced with a binary cross-entropy with logits loss; 5) each model was trained for a minimum/maximum of 30/80 epochs. If validation area under the receiver-operator curve (AUROC) did not increase for 5 consecutive epochs, training was terminated and the model with the highest AUROC was selected.

**Geometric deep learning framework:** To address outstanding issues in generalising individualised HCP_MMP1.0 parcellations to the lower quality UKB imaging data, a novel geometric deep learning framework was implemented on icospheric cortical surfaces (Fawaz et al., 2021). Each segmentation network (one per cortical area) was based on a U-Net (Ronneberger et al., 2015) with 5 encoding and 5 decoding blocks (Figure 2). Each block contained a single MoNet convolutional layer parameterised with K = 37 Gaussian kernels and polar pseudo-coordinates (Monti et al., 2017), followed by a SiLU nonlinearity (Elfwing et al., 2018). The optimal number of Gaussian kernels was determined by first assessing model performance for selected cortical areas using K = [5, 10, 15, 20, 25, 30, 35, 40, 45, 50] (optimal model performance when K = 35) and then further optimising by making unit

---

1. Generating UKB specific ICA maps for multimodal alignment was considered, but was ultimately decided against in order to minimise differences in preprocessing with the HCP. However, group ICA of 3000 UKB subjects at dimension = 40 generated very similar components to group ICA of the same dimensionality in the HCP.

2. These are the same features as for (Glasser et al., 2016a) with the exclusion of task MRI features (1 mean task fMRI activation map, and 20 task fMRI spatial ICA maps) which had no correlates in UKB

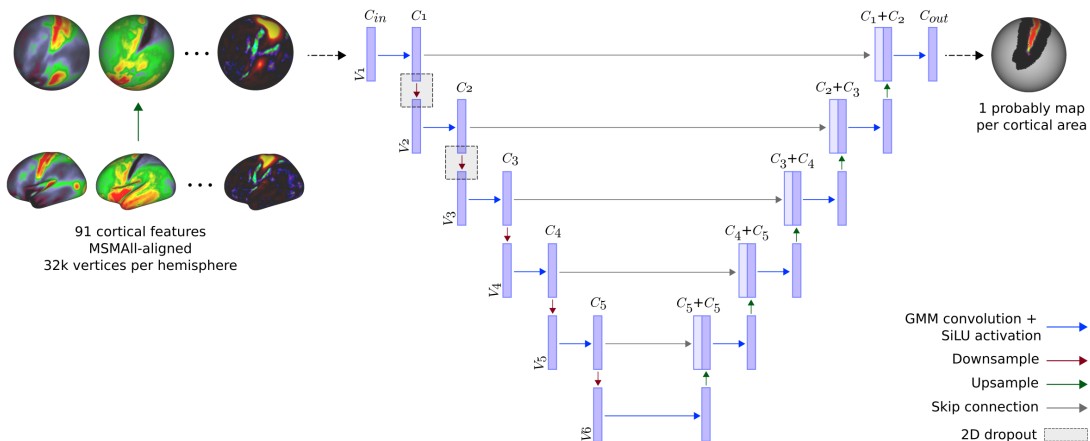

Figure 2: Geometric deep learning architecture. $C$ represents the number of channels, and $V$ represents the number of vertices. For the segmentation networks: $[C_{in}, C_1, C_2, C_3, C_4, C_5, C_{out}] = [91, 32, 64, 128, 256, 512, 1]$. For the reconstruction autoencoder, skip connections were removed and $C_{out}$ varied across cortical areas.

adjustments of K. Downsampling and upsampling were achieved following the procedure described in (Fawaz et al., 2021). Model overfitting was addressed by using 2D dropout ($p$=0.2) (Srivastava et al., 2014) prior to the first two downsamples. Training was carried out in two stages. During the first ('pretraining') stage, a reconstruction autoencoder was trained for the purpose of initialising the weights of the segmentation model for each cortical area. The reconstruction autoencoder had the same architecture as Figure 2 but lacked skip connections. Each reconstruction autoencoder was tasked with reconstructing only the most informative cortical features for a given area, as determined through visual inspection. All 91 cortical features were used as inputs in order to match the input weight dimensions for the segmentation model in the second training stage. However, the number of output channels (labels) to reconstruct was variable across cortical areas, depending on how informative each cortical feature was. The reconstruction autoencoder model weights were initialised with xavier initialisation (Glorot and Bengio, 2010). Segmentation was performed in the second training stage, using the pretrained model weights. Models inputs were the same 91 cortical features, whilst labels were subject-specific parcellations derived from (Glasser et al., 2016a), generated using the original FCNN architecture. Each segmentation model output was transformed into a probability map using a sigmoid function, where values represented the probability that each vertex belongs to a given cortical area. Final parcellations were generated by aggregating across all classifiers using a winner-takes-all approach, with each vertex being assigned the label with the highest probability.

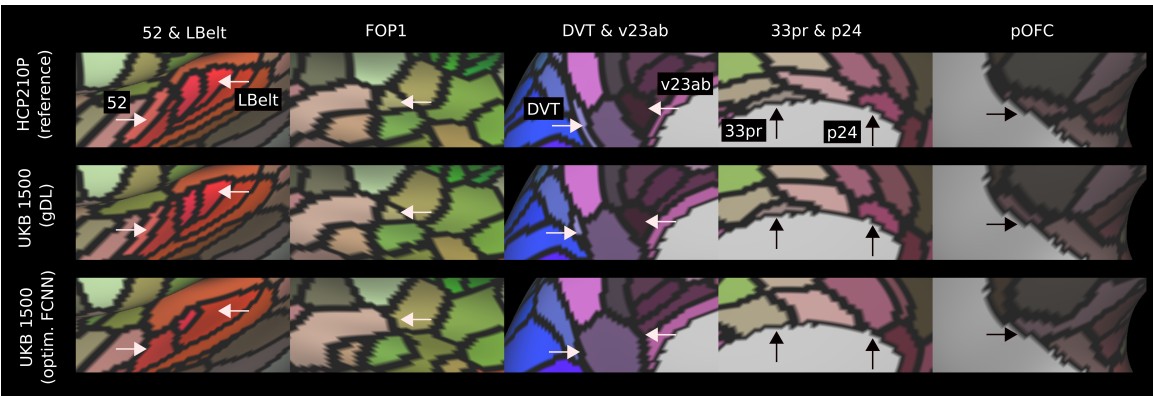

Figure 3: Comparison of selected cortical areas across gDL and optimised FCNN architectures in UKB. All cortical areas highlighted are absent from the UKB 1500 group average parcellation when using optimised FCNN architecture. However, all cortical areas are present in UKB 1500 group average parcellation using gDL, and share very similar topography to HCP210P group average. Only left hemisphere shown for visualisation purposes.

### 2.0.4. IMPLEMENTATION AND TRAINING

Models were trained and tested using the HCP210P and HCP29T subsets described in (Glasser et al., 2016a). Cortical features and labels were upsampled from a standard resolution (32,492 vertices) to a sixth order icospheric resolution (40,962 vertices). All cortical features were normalised within-subject and within-feature to a mean of 0 and standard deviation of 1, and extreme values were clipped at $\pm$ 4 standard deviations. Autoencoder models were trained and validated using the mean absolute error, whilst segmentation models were trained using a binary cross-entropy loss with logits, and validated using area under the receiver-operator curve. All models were trained with a batch size of 1, and optimisation was perfomed using AdamW (Loshchilov and Hutter, 2017) with a learning rate of $10^{-3}$. All models were trained on a single NVIDIA RTX 24GB GPU, for a minimum/maximum of 15/30 epochs. Model training was terminated if there were no improvements in validation performance after 5 consecutive epochs. Once trained, segmentation models were then applied to all UKB subjects. To account for differences in the intensity distributions across each population, all UKB subject features were histogram matched to the group average of the HCP 210P feature maps used for training. No UKB subjects were seen during training.

## 3. Results & Discussion

Figure 3 compares the group average parcellation of 1500 UKB subjects using the optimised FCNN and gDL frameworks. Only 345 (left hemisphere: 172; right hemisphere: 173) of 360 cortical areas were present in the optimised FCNN group average, whilst all cortical areas were present in the results from our proposed gDL method. Of note, the original HCP FCNN framework (Glasser et al., 2016a) was unable to generate any meaningful parcellations for

individual UKB data, most likely due to a combination of lower data quality in the UKB and the spatial independence of the FCNN. Moreover, using a single gDL network to parcellate all areas simultaneously resulted in a group average parcellation containing only 135 of 180 areas per hemisphere.

The proposed gDL model also outperformed the optimised FCNN architecture in average areal detection rate (99.1% vs. 88.7%) and lowest areal detection rate (80.9% vs. 29.9%) across both hemispheres. Importantly, these gDL results were not a result of each segmentation model predicting one consistent parcellation across subjects: UKB areal probability distributions were highly similar to those in the HCP, and demonstrate that cortical areas vary in location, even after areal feature-based cortical alignment (Figure 4a, third row). Moreover, visual inspection indicated that our method was sensitive to atypical cortical topographies, with 3.7% (n = 54) of subjects having split 55b and 1.7% (n=26) having shifted 55b (Figure 4b).

Both the optimised FCNN and gDL frameworks performed well in unseen HCP210V cortical data, and demonstrate improvements compared to the original HCP FCNN used in (Glasser et al., 2016a) (Table 1). The smaller differences in performance between the optimised FCNN and gDL frameworks are expected, given that the amount and quality of MRI data per subject in the HCP is substantially higher compared to the UKB.

Despite marked differences in imaging protocols, these results highlight the excellent replicability of the HCP_MMP1.0 parcellation using the proposed gDL framework (Figure 4a), which is pertinent given concerns about reproducibility in neuroimaging (Poldrack et al., 2017), and stands in contrast to the replicability of other fully data-driven cortical parcellations in independent datasets (Gordon et al., 2016; Lewis et al., 2022). However, an important limitation is that both optimised FCNN and gDL frameworks were trained using labels generated from the original HCP FCNN (Glasser et al., 2016a), which should not be considered ground truth. Further work is required to generate more accurate training labels, and to understand how these impact model performance and parcellation accuracy.

| Model | Original FCNN | Optimised FCNN | gDL |
|---|---|---|---|
| Average areal detection rate (%) | 96.4 | 99.3 | **99.9** |
| Lowest areal detection rate (%) | 70.3 | 89.6 | **96.4** |
| Split 55b (%) | 4.6 | **5.1** | 1.7 |
| Shifted 55b (%) | 4.4 | **5.6** | 2.9 |

Table 1: Comparison of model performance in HCP 210V validation set. FCNN: fully-connected neural network; gDL: geometric deep learning

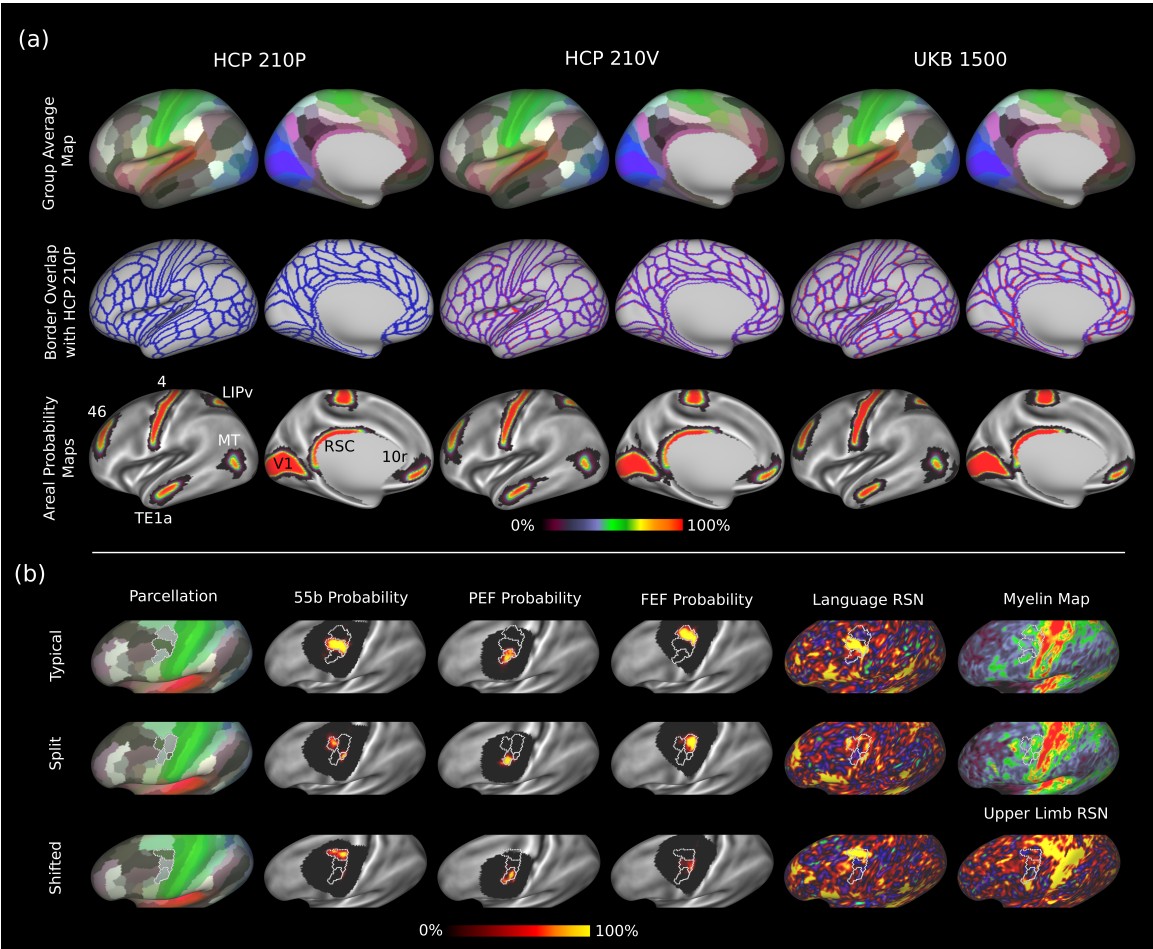

Figure 4: Multimodal parcellation in the HCP and UKB. (a) First row: group average maps for HCP 210P, HCP 210V and 3000 UKB subjects. Second row: Border overlap between HCP 210P (blue; seen during training), HCP 210V and UKB 3000 (red; not seen during training). Areas of purple indicate overlap between borders. The Dice overlap ratio between HCP 210P and UKB 3000 was 0.986, relative to 0.996 between HCP 210P and 210V group maps. Third row: Probabilistic maps of areas V1, 4, RSC, MT, LIPv, TE1a, 46, and 10r, overlaid on MSMAll-aligned S1200 HCP average curvature map. As expected, spatial variability of cortical areas between subjects changes across the cortex, and this variability is consistent between HCP 210P and HCP 210V reported in (Glasser et al., 2016a) using a fully connected neural network, and UKB 3000 using our method. (b) Representative examples of typical (first row), split (second row) and shifted (third row) area 55b topographies. Alongside each individual parcellation are the probability maps for areas 55b, frontal eye field (FEF) and premotor eyefield (PEF), and the language spatial ICA map (from ICA d=40). T1w/T2w maps are shown for typical and split 55b topographies, whilst the upper limb spatial ICA map is shown for the shifted 55b topography (Glasser et al., 2016a). All maps are displayed on an inflated cortical surface. Results were similar for both hemispheres, but only the left hemisphere is shown.

## 4. Conclusion

Here we present and evaluate an approach for generating individualised multimodal cortical parcellations that generalise to markedly different datasets, and highlights the reproducibility of the HCP_MMP1.0 parcellation in independent cohorts.

## Acknowledgments

Data were provided [in part] by the Human Connectome Project, WU-Minn Consortium (Principal Investigators: David Van Essen and Kamil Ugurbil; 1U54MH091657) funded by the 16 NIH Institutes and Centers that support the NIH Blueprint for Neuroscience Research; and by the McDonnell Center for Systems Neuroscience at Washington University. The HCP was approved by the internal review board of Washington University in St. Louis (IRB #201204036). Data can be downloaded from https://db.humanconnectome.org/app/template/Login.vm. UKB data were accessed through application number 53775, under PI Emma C. Robinson. The UKB was approved by the National Information Governance Board for Health and Social Care and the National Health Service North West Centre for Research Ethics Committee (Ref: 11/NW/0382). Application to access to UKB data can be found at https://www.ukbiobank.ac.uk/enable-your-research/apply-for-access.

L.Z.J.W. is funded by the Commonwealth Scholarship Commission, United Kingdom. M.F.G. and D.C.V.E are supported by National Institutes of Health, USA (grant number MH060974). F.A.A. is funded by the UK Medical Research Council and the Wellcome Trust. Y.G. is supported by the King's-China Scholarship Council. S.D. is supported by the EPSRC Centre for Doctoral Training in Smart Medical Imaging (EP/S022104/1). E.C.R. is supported by an Academy of Medical Sciences/the British Heart Foundation/the Government Department of Business, Energy and Industrial Strategy/the Wellcome Trust Springboard Award (SBF003/1116) and MRC Methodology grant (MR/V03832X/1). E.C.R., R. B., and S.M.S. are supported by a Wellcome Trust Collaborative Award (215573/Z/19/Z). The authors acknowledge use of the King's Computational Research, Engineering and Technology Environment (CREATE) (https://doi.org/10.18742/rnvf-m076).

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
