# OpenReview forum: "Generalising the HCP multimodal cortical parcellation to UK Biobank using geometric deep learning"
_MIDL.io/2026/Conference — MIDL 2026 Poster_

### Official Review · Reviewer_CVh7 · 2025-12-24

**Confidence:** 5
**Preliminary Rating:** 4
**Final Rating:** 4

**Summary:**

This paper describes a practical solution to improve existing FCNN implementation to generate individualized parcellation on both HCP and UK biobank data. Although the innovation in machine learning methods is relatively limited, the presented approach could have broad applications on fMRI analysis.

**Strengths:**

1. Clear presentation of methods, informative visualizations.

2. HCP and UK-biobank are by far the two largest public fMRI datasets. A generalizable parcellation method on both dataset has the potential application to integrate two datasets for brain function analysis.

3. The proposed approach shows significant improvement on visualizations and Table 1.

**Weaknesses:**

1. The training scheme could be further improved. Batch size=1 is a uncommon choice in machine learning, would be helpful to include ablation on batch size and other training parameters.

2. Performance on UK biobank data lacks quantitative results like Table 1 HCP 210V.

3. There might not be enough time to do this in rebuttal, but since the objective of the method is to develop individualized parcellation on two public datasets, it would be useful to run the training on the entire HCP and UK biobank data and then share the trained weights or resulting parcellations. If labels are not all available, some pseudo label training approach could be employed.

**Detailed Comments:**

see weakness

**Justification Of Final Rating:**

No rebuttal is received from the author. I am keeping my original rating as weak accept, primarily based on the application value of this paper in fMRI analysis for joint utilization of HCP and UK biobank dataset.

**Justification Of The Preliminary Rating:**

The application value of the paper overwhelms its deficiency in presentation and methodological details. A practical way to generate individualized parcellation on HCP and UK biobank dataset would be significant for the field of fMRI analysis.

**Questions To Address In The Rebuttal:**

see weakness

---

### Official Review · Reviewer_ib9i · 2025-12-31

**Confidence:** 4
**Preliminary Rating:** 4
**Final Rating:** 4

**Summary:**

This paper proposes a geometric deep learning (gDL) framework using MoNet-based U-Nets to adapt HCP cortical parcellations to the substantially sparser imaging protocols of the UK Biobank. To bridge the domain gap, the method uses a two-stage training scheme with autoencoder pretraining followed by segmentation training. Results show higher areal detection rates than an optimized FCNN baseline and suggest sensitivity to individual topographic variability

**Strengths:**

- The paper tackles the practical challenge of transferring HCP-style cortical parcellation to UK Biobank data despite substantial differences in imaging protocols, which is a relevant and important problem for large neuroimaging studies.
- Replacing vertex-wise FCNNs with a spatially aware surface-based gDL framework is a reasonable design choice for cortical labeling on meshes.
- The proposed method achieves higher areal detection rates than an optimized FCNN baseline on UKB, particularly for small or variable regions that are often missed.
- The model is able to capture subject-specific differences, such as split or spatially shifted instances of area 55b that persist even after multimodal alignment, indicating preserved biological sensitivity under lower-quality data.

**Weaknesses:**

- The models are trained on parcellation labels generated by the original HCP FCNN pipeline, which the authors explicitly state should not be considered ground truth. As a result, the proposed method may inherit or amplify biases from the earlier model rather than improving true anatomical accuracy.
- Quantitative evaluation relies mainly on areal detection rate, with several key conclusions supported by visual inspection, making it difficult to assess segmentation accuracy, boundary quality, or failure modes in UKB.
- The framework requires training 360 separate binary classifiers, one per cortical area, which is computationally expensive and less scalable than unified multi-class surface segmentation models. Although a single gDL network was explored, it performed poorly, and the trade-off between efficiency and performance is not well analyzed.
- Comparisons are largely restricted to an optimized FCNN baseline, and autoencoder pretraining relies on feature selection based on visual inspection, making it difficult to clearly attribute the observed gains to the proposed gDL design.

**Detailed Comments:**

- It would be helpful to clarify the definition of areal detection rate, including how area presence is determined, any thresholds used, and the role of group averaging.
- Autoencoder pretraining relies on reconstructing most informative features selected by visual inspection. Providing a brief description of the selected features or a rule-based criterion would improve reproducibility.
- Given the substantial age difference between the HCP and UKB cohorts, a short discussion on whether histogram matching might remove biologically meaningful age-related differences would be useful.
- Since the approach trains 360 area-specific networks, reporting total training time and inference cost would help readers assess its practical feasibility.

**Justification Of Final Rating:**

The authors did not address my concerns in the rebuttal, and my overall assessment therefore remains unchanged. I will keep my original score.
The authors did not address my concerns in the rebuttal, and my overall assessment therefore remains unchanged. I will keep my original score.

**Justification Of The Preliminary Rating:**

The paper studies the problem of applying HCP-style cortical parcellation to lower-resolution UK Biobank data and proposes a surface-based geometric deep learning approach that improves areal detection compared to an optimized FCNN baseline. However, the evaluation relies on FCNN-derived labels, a relatively limited set of validation metrics, and a narrow range of baseline comparisons. In addition, the requirement to train area-specific models raises questions about architectural efficiency and scalability.

**Questions To Address In The Rebuttal:**

- Since training labels are derived from the original HCP FCNN pipeline, could the authors clarify what evidence supports that higher areal detection rates correspond to improved anatomical validity, rather than simply producing denser or more confident predictions in UKB?
- How sensitive are the results to the choice of the 30 mm geodesic searchlight radius? Have the authors explored whether different searchlight sizes affect detection performance?
- The single gDL model detected substantially fewer areas (only 135/180 areas) than the per-area classifiers. Could the authors comment on why the unified model underperforms, and whether this is mainly due to class-imbalance of small cortical areas, or architectural limitations?
- Beyond areal detection rate, could the authors provide additional UKB-side validation, even limited, such as boundary consistency, uncertainty estimates, or comparisons to an alternative reference?

---

### Official Review · Reviewer_mzmM · 2026-01-07

**Confidence:** 4
**Preliminary Rating:** 3
**Final Rating:** 3

**Summary:**

The authors propose a geometric deep learning framework to generate individualised HCP parcellations and evaluated on UK Biobank Dataset. They compared with the original FCNN approach and replaced it with a surface-based UNET approach. They pretrained using an autoencoder to improve the parcellation results in lower-quality UK Biobank data. The authors report increased improved average detection rate across all areas.

**Strengths:**

1. The motivation is very clear and has practical value. This work directly tackles one of the bottlenecks in fMRI research, where parcellations from higher-quality HCP to lower-quality datasets like the UK Biobank are difficult.
2. The choice of using a geometric surface-based approach addresses the limitations of previous FCNN, where the neighboring vertices' shared information is explicitly modeled and used as a strong geometric prior.
3. The proposed method showed substantial improvement in the detection rate. Compared with the optimized FCNN baseline, gDL showed over 10% average detection rate increase while reducing missing area failures.

**Weaknesses:**

1. Limited methodological novelty. The proposed approach is largely a straightforward surface-based U-Net architecture plus a standard two-stage pretraining strategy.

2. The ground truth labels used are generated by the original HCP FCNN, which makes it hard to interpret the authors' claim for the more accurate parcellation results.

3. The evaluation on UK Biobank data required UKB inputs to be aligned and histogram-matched to the feature distributions in HCP. The authors did not show ablation studies to validate the robustness of all the harmonizations.

**Detailed Comments:**

1. There is a small typo in the Fully-connected framework section, where the second improvement is written as "tahn" and it should be tanh in the description

**Justification Of Final Rating:**

The authors propose a geometric deep learning framework to generate individualised HCP parcellations and evaluated on UK Biobank Dataset. The authors did not address my concerns during the rebuttal, and I cannot verify whether the limitations can be addressed. I therefore keep my original score and rationale.

**Justification Of The Preliminary Rating:**

The methodology presented by the authors is simple but has strong clinical and clear motivation. The reported performance improvement on the UK Biobank shows the effectiveness of generalizing HCP individualized parcelattions. However, the current model is trained on labels generated by the original HCP FCNN and the authors only showed evaluation on the detection rate, which may not capture boundary precision, topological correctness.

**Questions To Address In The Rebuttal:**

1. What is the probability threshold used for the winner-takes-all approach?  Could you clarify if any other post-processing was used?

2. Could you also clarify the cortical feature choices?

3. How much of the UKB parcellation performance improvement is dependent on the ICA alignment or histogram matching? Could you show some ablation results on this?

---

### Meta-Review · Area_Chair_sTNz · 2026-02-09

**Recommendation:** Accept (Poster)
**Confidence:** 4

**Metareview:**

This paper addresses an important and practical problem, that of generalizing HCP-style multimodal cortical parcellations to UK Biobank, using a well-motivated geometric deep learning framework that yields clear, substantial gains in areal detection and preserves individual variability. While methodological novelty is incremental, the application value and demonstrated improvements justify acceptance.

---

### Decision · Program_Chairs · 2026-02-13

Accept (Poster)